# Weighting What Matters: Boosting Sample Efficiency in Medical Report Generation via Token Reweighting

**Alexander Weers**[1,2]                                    ALEXANDER.WEERS@TUM.DE

**Daniel Rueckert**[1,2,3]                                 DANIEL.RUECKERT@TUM.DE

**Martin J. Menten**[1,2]                                  MARTIN.MENTEN@TUM.DE

[1] *TUM School of Computation, Information and Technology, Technical University of Munich, DE*

[2] *Munich Center for Machine Learning (MCML), DE*

[3] *Department of Computing, Imperial College London, UK*

## Abstract

Training vision-language models (VLMs) for medical report generation is often hindered by the scarcity of high-quality annotated data. This work evaluates the use of a weighted loss function to improve data efficiency. Compared to standard cross-entropy loss, which treats all token prediction errors equally, the reweighted loss shifts the focus to semantically salient tokens with outsized clinical importance. In experiments on ophthalmological report generation, we show that this simple method improves efficiency across multiple data scales, achieving similar report quality with up to ten times less training data.

**Keywords:** Vision-Language Models, Sample Efficiency, Ophthalmology

## 1. Introduction

Automated generation of medical reports via vision-language models (VLMs) has the potential to standardize diagnostic workflows, reduce clinician workload, and lower healthcare costs (Hartsock and Rasool, 2024). However, unlike general-domain VLMs, medical models are commonly trained in data-constrained settings, where paired image-report datasets are small and expensive to curate, particularly in smaller medical subdisciplines such as ophthalmology (Holland et al., 2025).

In order to maximize the efficiency of limited data, this work builds upon the observation that not all words in a medical report carry the same clinical relevance. Standard language model training with a per-token cross-entropy loss treats all token prediction errors equally, even though this may not reflect their semantic importance. For instance, predicting "OCT image" instead of "OCT scan" is largely inconsequential, whereas confusing "multiple drusen" with "no drusen" changes the clinical meaning of the report. We postulate that training objectives should reflect the semantic and clinical importance of tokens rather than assuming all words are equally important.

This idea connects to prior work on reweighting losses. Focal loss (Lin et al., 2017) can be understood as a general form of weighted cross-entropy that emphasizes harder examples. Related approaches have been used for the training of medical language models, assigning larger weights to clinically relevant terms by upweighting UMLS-matched terminology (Wu et al., 2025) or predefined word sets (Drago et al., 2025). However, it remains unclear how effective such token reweighting is for data-constrained report generation and how it depends on dataset size and the choice of upweighted terms. In this work, we perform a systematic analysis of token reweighting across various data set sizes and compare the effects of different clinically motivated keyword sets on the quality of generated reports.

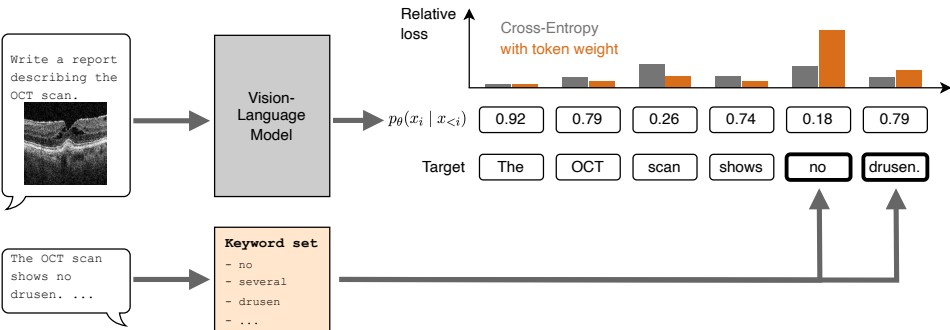

Figure 1: By upweighting tokens from a pre-defined set of clinical keywords, the model increases the penalty for clinically significant errors relative to non-diagnostic tokens, and ultimately prioritizes correct prediction of semantically important terms.

## 2. Method

To improve ophthalmologic report generation, we define three sets of keywords that we deem most relevant with regard to the semantic content. Quantitative keywords $\kappa_Q$ (n=34) describe the severity or extent of a finding, for example *thick*, *several*, or *increased*. Diagnostic keywords $\kappa_D$ (n=22) correspond to biomarkers and disease-relevant concepts, such as *healthy*, *late*, *fluid*, and *drusen*. We evaluate the benefit of upweighting these sets both individually, as well as a combined set $\kappa_C$ (n=56). The full list of keywords is provided in Appendix A.

For a tokenized target sequence $x = (x_1, \ldots, x_T)$, we identify all keywords from $\kappa$ in the reference report and map them to token indices using the model's tokenizer. If a matched word or phrase is split into multiple subword tokens, all corresponding token indices are selected. This yields a set of clinically significant token positions $I_\kappa(x) \subseteq \{1, \ldots, T\}$. Subsequently, the standard token-level cross-entropy is replaced with the normalized weighted objective

$$\mathcal{L}_{\text{tw}}(x, \lambda) = -\frac{1}{\Lambda} \sum_{i=1}^{T} \lambda_i \log p_\theta(x_i \mid x_{<i}), \qquad \Lambda = \sum_{i=1}^{T} \lambda_i, \qquad \lambda_i = \begin{cases} \gamma, & \text{if } i \in I_\kappa(x), \\ 1, & \text{otherwise,} \end{cases}$$

where $\gamma > 1$ is the upweighting factor for clinically important tokens and the normalization by $\Lambda$ maintains the overall loss scale across reports with different numbers of keywords.

## 3. Experiments & Results

In experiments, we quantify the benefit of token reweighting for generation of ophthalmological reports. We use a vision-language model consisting of a pretrained image encoder (Holland et al., 2024), a Llama3-3B language model (Grattafiori et al., 2024), and a connecting projector layer (Zhu et al., 2024). During fine-tuning using visual question-answer pairs (see Appendix B), we freeze the image encoder and update the projector layer as well as the language model via LoRA (Hu et al., 2022). In each case, we compare model training with and without token reweighting. To ensure robust results we conduct a hyperparameter sweep

| Data | Method | AMD | Biomarker |
|------|--------|-----|-----------|
| 1% | standard | 0.308 | **0.232** |
|    | weighted | **0.348** | 0.230 |
| 3% | standard | 0.337 | 0.289 |
|    | weighted | **0.358** | **0.304** |
| 10% | standard | 0.422 | 0.329 |
|     | weighted | **0.490** | **0.348** |
| 30% | standard | 0.450 | 0.336 |
|     | weighted | **0.483** | **0.413** |
| 100% | standard | 0.481 | 0.385 |
|      | weighted | **0.573** | **0.429** |

Table 1: $F1_{macro}$ performance of standard vs. keyword-weighted cross-entropy across varying dataset sizes.

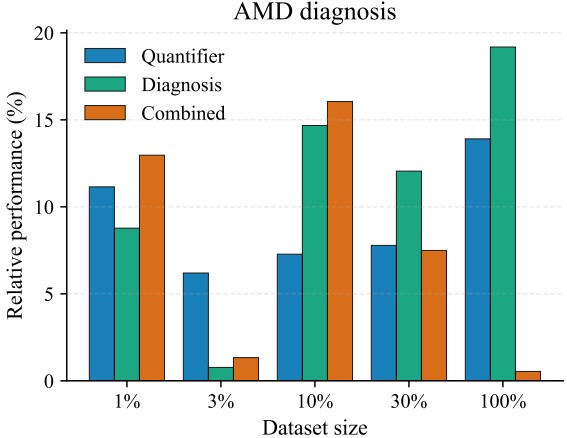

Figure 2: Relative performance gain on AMD classification of different keyword sets compared to the unweighted baseline.

over several learning rates and token weight factors $\gamma$ within a four-fold cross-validation setup (see Appendix C). The best-performing models were evaluated on a held-out test set. We report the mean $F1_{macro}$ scores of the reports with regard to age-related macular degeneration (AMD) staging and biomarker identification.

Token weighting consistently outperforms standard cross-entropy across nearly all data regimes and metrics (see Table 1). Notably, token weighting significantly improves sample efficiency. At many dataset sizes using a weighted loss had a stronger benefit than increasing the dataset size by a factor of three. For AMD staging, models trained on only 10% of the dataset using a weighted loss outperform the unweighted baseline trained on the full dataset.

All evaluated keyword sets boost performance across all dataset sizes (see Figure 2). The diagnostic keyword set is particularly effective, highlighting the high relevance of biomarker and disease-relevant concepts in medical report generation.

## 4. Conclusion

This work has shown that upweighting a curated set of clinically relevant keywords during VLM training improves the model's ability to generate ophthalmological reports. Across various dataset scales, using a weighted loss formulation improved the quality of generated reports with regard to AMD staging and biomarker identification accuracy, in many cases outperforming standard model training with three times more data. We believe this strategy can function as a simple, effective tool for data-efficient training of vision-language models in medical domains with limited annotated data.

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

## Appendix A. Keyword sets

We use the following keyword sets:

### Diagnostic keywords

| | | |
|---|---|---|
| healthy | inactive | subretinal |
| normal | hyperreflective | fluid |
| early | hyporeflective | |
| intermediate | drusen | atrophy |
| late | drusenoid | atrophic |
| wet | elevation | |
| dry | irregularity | transmission |
| active | intraretinal | hypertransmission |

### Quantitative keywords

| | | | |
|---|---|---|---|
| yes | decreased | medium | smaller |
| no | one | moderate | larger |
| small | two | moderately | |
| large | some | advanced | largest |
| thin | several | extensive | thinned |
| thick | multiple | very | thinning |
| increase | many | significant | |
| increased | minimal | thickened | slight |
| decrease | slightly | thickening | significantly |

## Appendix B. Dataset

We use the dataset and preprocessing approach introduced in (Holland et al., 2025). Specifically, we create a visual question-answer dataset from tabular information for 41,926 OCT scans, and detailed reports for 295 OCT scans. This results in a VQA dataset of 915,229 samples. For model evaluation, we use held-out test data consisting of 86 detailed reports, from which AMD stage and biomarker information (present/absent) are extracted.

## Appendix C. Hyperparameter details

For every configuration (unweighted or reweighted), we perform a hyperparameter sweep over a set of learning rates $\alpha \in \{6.5\mathrm{e}{-5}, 1\mathrm{e}{-4}, 2.15\mathrm{e}{-4}, 6.5\mathrm{e}{-4}\}$ and for the weighted runs over a set of weighting factors $\gamma \in \{2.0, 3.5, 6.0\}$. For every run we evaluate the $F1_{\mathrm{macro}}$ performance on AMD staging and biomarker detection. For AMD staging we consider the classes *healty*, *early/intermediate*, *late wet*, and *late dry*. For biomarker detection we extract the presence or absence of *drusen*, *retinal pigment epithelium*, *pigment epithelial detachment*, *hyperreflective foci*, *hypertransmission*, *fibrosis*, *subretinal fluid*, and *intraretinal fluid* and calculate the $F1_{\mathrm{macro}}$ score. Final model selection is based on both scores with equal importance.

