# OpenReview forum: "Weighting What Matters: Boosting Sample Efficiency in Medical Report Generation via Token Reweighting"
_MIDL.io/2026/Short_Papers — MIDL 2026 - Short Papers Poster_

### Official Review · Reviewer_E1Co · 2026-05-02

**Rating:** 4
**Confidence:** 4

**Review:**

This paper provides a clear and impactful solution to the challenge of data scarcity in medical AI by acknowledging that clinical accuracy depends on a small subset of high-stakes terms. The methodology is technically sound and highly practical, offering a "simple, effective tool" that can be integrated into existing VLM pipelines without architectural changes. The systematic evaluation across varying dataset sizes (from 1% to 100%) and the use of distinct keyword categories provide strong empirical evidence for the approach's robustness. While the reliance on a predefined keyword set might be seen as a limitation, the authors demonstrate that even small, expertly curated lists yield outsized benefits for clinical utility

**Summary:**

The authors propose a weighted loss function that shifts the training focus from common linguistic tokens to semantically critical clinical keywords. Using a curated set of 56 keywords—categorized into quantitative (e.g., "thick," "several") and diagnostic (e.g., "drusen," "fluid") terms—the study replaces standard cross-entropy with a normalized weighted objective. Experiments utilizing a Llama3-3B backbone and ophthalmological OCT scans show that the reweighted loss consistently outperforms the standard approach across all dataset scales, particularly in tasks like AMD staging and biomarker identification. Notably, the diagnostic keyword set proved most effective, and the strategy allowed a model trained on 10% of the data to surpass an unweighted model trained on the full dataset.

**Strengths:**

The primary strength of the work is the empirical proof that clinical domain knowledge, encoded as simple token weights, can drastically improve the learning curve of large-scale VLMs. The finding that diagnostic keywords are more impactful than quantitative ones offers valuable guidance for future medical VLM development

**Weaknesses:**

The study is limited by its focus on a single imaging modality (OCT) and a specific pathology (AMD). Additionally, while the hyperparameter sweep for the upweighting factor $\gamma$ is mentioned, a more detailed analysis of the sensitivity of the model to different $\gamma$ values across various tasks would further strengthen the results

**Justification Of Rating:**

The paper addresses a significant bottleneck in medical imaging—data scarcity—with a robust, easily reproducible method. The results are compelling and show a clear clinical benefit, making it a strong contribution to the MIDL short paper track.

---

### Decision · Program_Chairs · 2026-05-08

Accept (Poster)